# DIFFERENTIABLE ARCHITECTURE COMPRESSION

## ABSTRACT

In many learning situations, resources at inference time are much more constrained than resources at training time. This paper studies a general paradigm, called Differentiable ARchitecture Compression (DARC), that combines model compression and architecture search to learn models that are resource-efficient at inference time. Given a resource-intensive base architecture, DARC utilizes the training data to learn which sub-components can be replaced by cheaper alternatives. The high-level technique can be applied to any neural architecture, and we report experiments on state-of-the-art convolutional neural networks for image classification. For a WideResNet with 97.2% accuracy on CIFAR-10, we improve single-sample inference speed by $2.28\times$ and memory footprint by $5.64\times$, with no accuracy loss. For a ResNet with 79.15% Top1 accuracy on ImageNet, we improve batch inference speed by $1.29\times$ and memory footprint by $3.57\times$ with 1% accuracy loss. We also give theoretical Rademacher complexity bounds in simplified cases, showing how DARC avoids overfitting despite over-parameterization.

## 1 INTRODUCTION

In machine learning, resources at inference time are often much more constrained than at training time. For example, while neural networks for computer vision and natural language processing (NLP) are routinely trained using GPUs, trained networks are often deployed on embedded systems or mobile devices with limited memory and computational power. As another example, it is common to train a model that will be applied continuously in production; while training occurs for a limited time, the machine performing inference may run indefinitely, and so learning a more efficient model can directly reduce costs associated with hardware or energy usage. As a result, many recent papers have studied deep model compression and acceleration. Most of these papers provide resource efficient model components (Jaderberg et al., 2014; Zhang et al., 2016; Wu et al., 2017) or methods to prune or quantize parameters (LeCun et al., 1990; Polyak & Wolf, 2015; Li et al., 2016a; He et al., 2017; Luo & Wu, 2017; Luo et al., 2017; Zhuang et al., 2018).

We propose a general paradigm called Differentiable ARchitecture Compression (DARC) for learning in the context of constrained resources at inference-time. Rather then suggesting a specific cheap component and using it blindly throughout a neural network, or trying to tune layer hyperparameters as in network pruning or quantization, our approach is inspired by Neural Architecture Search (NAS) (Zoph & Le, 2016; Liu et al., 2018a; Pham et al., 2018; Kandasamy et al., 2018; Liu et al., 2018b). DARC starts with a resource-intensive network design and uses training data to learn which components can be replaced with efficient alternatives, while maintaining model output quality. The resource requirement of the final model is controlled via a regularization term, which can be flexibly defined depending on the objective, such as minimizing inference time or memory footprint.

DARC has a clear intuitive advantage when compared to methods that quantize or prune parameters; these approaches are inherently restricted in their search space. They cannot replace a layer with a structurally different layer, or replace two or more layers with a shallow alternative. As examples, it might be the case that a convolutional layer cannot be pruned without hurting performance but can be replaced with a depthwise-separable convolution, or an LSTM layer might not be amenable to weight pruning, but could be replaced with a more efficient self-attention layer. DARC, applied to deep networks, offers a way to search a rich space in the context of model compression.

The high-level idea is to partition the network into components, and explore alternatives to these components *simultaneously*. Replacing all components simultaneously is crucial, as it provides a data-driven way to decide *which* components can be replaced with cheap alternatives. Indeed, replacing layers blindly may sacrifice too much prediction performance, as can be seen in Mo-

bileNets (Howard et al., 2017; Sandler et al., 2018), which exclusively use depthwise-separable convolutions; although computationally efficient, even the most accurate MobileNet model is far less accurate at ImageNet classification than, say, ResNet50.

We frame the problem of learning good replacement components as a sparse ensemble learning problem. This view allows us to draw guidelines from simpler analyzable cases, leading to a simple, gradient-based learning scheme that avoids over-fitting and is fast enough to apply DARC directly to large datasets such as ImageNet. This contrasts from most NAS methods, which first learn an architecture on a small dataset and then fit this architecture on the larger dataset.

We present experiments on networks commonly used in computer vision. By applying our techniques on the ResNet Architecture (He et al., 2016), we outperform state-of-the-art models on both ImageNet and CIFAR-10 datasets, in terms of accuracy vs. throughput and accuracy vs. model size. A few results from our compression framework: for a WideResNet model achieving $97.2\%$ accuracy on CIFAR-10, we improve single-sample inference speed by $2.28\times$ and memory footprint by $5.64\times$, with no loss in accuracy. For a modified ResNet50 model with $79.15\%$ Top1 accuracy on ImageNet, we improve inference speed by $1.29\times$ and memory footprint by $3.57\times$ with $1\%$ loss in accuracy. Both base models are publicly available from the GluonCV Model Zoo (Mod, 2018). Our experiments empirically demonstrate an intuitive observation that 'you get what you optimize for', in that models minimizing model size tend to be quite different from those maximizing throughput.

We note that, while our experiments are limited to image classification, DARC is applicable to any deep learning architecture, including models with recurrent cells or transformers, or indeed any sufficiently modular learning algorithm, as described in Section 2, and any task with a well-defined objective function in which we would like to reduce inference costs. We see this work as a proof-of-concept for capabilities of the DARC framework, and the results in this paper give a strong indication that DARC can be applied to NLP architectures, or optimized for metrics other than those we try here (e.g. for latency on devices other than GPU, or energy consumption).

## 2 GENERAL SETTING AND DARC ESTIMATOR

The intuition and motivation for our method starts with a task of model selection. Given a task and $J$ function families corresponding to candidate models, we are interested in finding the best model type for the task. We relax this combinatorial optimization problem to a more tractable differentiable optimization problem (whence the name "differentiable architecture compression"), by allowing convex combinations of these candidates. This can be thought of as a constrained form of ensemble learning in which weights are restricted to represent a convex combinations of individual learners.

Then, we posit a budget constraint: each model type has an associated cost (e.g., memory consumption or latency), and the overall cost of the ensemble is the sum of costs of the used models. Our task is then learn a convex ensemble over a subset of candidates, with total cost within budget. We now formalize this approach, with modifications to address technical challenges as they arise.

In the sequel, for any positive integer $J$, $[J] = \{1, 2, ..., J\}$ denotes the set of positive integers at most $J$, and $\Delta^J := \left\{ x \in [0,1]^J : \sum_{j \in [J]} x_j = 1 \right\}$ denotes the probability simplex over $J$ elements.

Consider the conventional supervised learning setting, in which we have an i.i.d. training dataset $(X_1, Y_1), ..., (X_n, Y_n) \overset{IID}{\sim} P_{(X,Y)}$ from some joint distribution $P_{(X,Y)}$ on $\mathcal{X} \times \mathcal{Y}$. Fix a loss function $\mathcal{L} : \mathbb{R} \times \mathcal{Y} \to [0, \infty]$, and a hypothesis class $\mathcal{H}$ of $\mathbb{R}$-valued functions. We would like to learn a function $h : \mathcal{X} \to \mathbb{R}$, $h \in \mathcal{H}$ that minimizes the risk $R(h) := \mathbb{E}_{(X,Y) \sim P_{(X,Y)}}[\mathcal{L}(h(X), Y)]$. The usual empirical risk minimization (ERM) estimator is $\widehat{h}_{\text{ERM}} := \arg\min_{h \in \mathcal{H}} \widehat{R}(h)$ where, for any hypothesis $h \in \mathcal{H}$, $\widehat{R}(h) := \frac{1}{n} \sum_{i=1}^{n} \mathcal{L}(h(X_i), Y_i)$ denotes the empirical risk. To derive our resource-constrained objective, we impose a few structural assumptions on our hypothesis class $\mathcal{H}$:

**(A1)** $\mathcal{H} = \text{Conv} \bigcup_{j \in [J]} \mathcal{H}_j$ is the convex hull of a union of $J$ classes $\mathcal{H}_1, ..., \mathcal{H}_J$.

**(A2)** Each class $\mathcal{H}_j$ class has a known cost $C_j \geq 0$ of using a hypothesis $h_j \in \mathcal{H}_j$ at test-time.

**(A3)** Costs are additive: hypothesis $h = \sum_{j=1}^{J} \alpha_j h_j \in \mathcal{H}$ has cost $C_{\ell_0}(\alpha) = \sum_{j=1}^{J} C_j 1_{\{\alpha_j > 0\}}$.

**(A4)** We have a known budget $B \geq 0$ for the final model at test-time.

As we show in Section 3, these assumptions arise naturally in architecture compression. Given assumptions equation A1-equation A4, the constrained ERM estimate is $\widehat{g} = \sum_{j \in [J]} \widehat{\alpha}_j \widehat{h}_j$, where

$$(\widehat{\alpha}_0, \widehat{h}_0) := \underset{\alpha \in \Delta^J, \widehat{h}_j \in \mathcal{H}_j}{\arg \min} \ \widehat{R} \left( \sum_{j \in [J]} \alpha_j h_j \right), \quad \text{subject to} \quad C_{\ell_0}(\alpha) \leq B. \tag{1}$$

The above estimator is difficult (NP-hard) to compute, due to the non-smooth, non-convex budget constraint $C_{\ell_0}(\alpha) \leq B$. Since this constraint bounds the $\ell_0$ norm of $\alpha$ (weighted by $C$), the usual remedy would be to relax the constraint to one on the $\ell_1$ norm of $\alpha$ (weighted by $C$), namely $C_{\ell_1}(\alpha) := \sum_{j \in [J]} C_j \alpha_j \leq B$. Unfortunately, due to the constraint that $\alpha$ lies in the probability simplex $\Delta^J$ (which implies $\sum_{j \in [J]} \alpha_j = 1$), the $\ell_1$ constraint is insufficient to induce sparsity on $\alpha$. Fortunately, sparse optimization on $\Delta^J$ is well-studied, with many solutions proposed (Pilanci et al., 2012; Kyrillidis et al., 2013; Li et al., 2016b). Due to ease of implementation, we adopt a simple but effective solution proposed by (Kyrillidis et al., 2013), which involves alternating gradient updates with a projection operation $P_{\Delta^J} : \mathbb{R}^J \backslash \mathbb{R}^J_- \to \Delta_J$, given by $P_{\Delta^J}(\alpha) = \frac{\alpha_+}{\|\alpha_+\|_1}$, where $\alpha_+ = (\max\{0, \alpha_1\}, ..., \max\{0, \alpha_j\}) \in \mathbb{R}^J_+$.

$P_{\Delta^J}$ is easy to compute, enforces the simplex constraint $\alpha \in \Delta^J$ exactly, and induces sparsity on $\alpha$. For an intuition of how this works, one can note that $\nabla \|\alpha\|_2^2/2 = \alpha$, so that the update $\alpha/\|\alpha\|_1 = \alpha - (1 - 1/\|\alpha\|_1) \alpha$ can be viewed as a gradient step for minimizing $-\|\alpha\|_2^2/2$ with adaptive step size $(1 - 1/\|\alpha\|_1)$. As a technicality, we note that the projection $P_{\Delta^J}(\alpha)$ is undefined when $\alpha \in \mathbb{R}^J_-$ has no positive components. However, for realistic gradient step sizes $\eta$, this never occurs, since, after each gradient update, $\sum_{j \in [J]} \alpha_j \geq 1 - O(\eta)$.

Finally, a natural initial point for our procedure is one where $C_{\ell_1}(\alpha) > B$, hence we re-express the constraint $C_{\ell_1}(\alpha) \leq B$ as a penalty $\lambda C_{\ell_1}(\alpha)$. Since the value of $\lambda$ corresponding to $B$ is not known *a priori*, we iteratively increase $\lambda$ until the solution of the optimization problem satisfies the budget constraint. The resulting DARC procedure is shown Algorithm 1. We note that the "stopping criterion" for the inner loop can be as simple as a fixed number of training epochs (as in our experiments), or a more sophisticated early-stopping criterion.

---

**Algorithm 1:** DARC algorithm for general hypotheses

---

**Data:** Training Data $\{(X_i, Y_i)\}_{i=1}^n$, $J$ candidates $h_{1,w_1}, ..., h_{j,w_j}$ with initial parameters $w_1, ..., w_j$ and costs $c_1, ..., c_j \geq 0$, initial cost penalty parameter $\lambda_0 > 0$, budget $B$.

**Result:** $\alpha, w_1, ..., w_j$ such that $h = \sum_{j \in [J]} \alpha_j h_{j,w_j}$ has small risk $R(h)$ and cost $C_{\ell_0}(\alpha) \leq B$

1  $\alpha \leftarrow (1/J, ..., 1/J)$, $\lambda \leftarrow \lambda_0$ ;
2  **while** $C_{\ell_0}(\alpha) > B$ **do**
3      **while** *stopping criterion is not met* **do**
4          $(\alpha, w_1, ..., w_J) \leftarrow (\alpha, w_1, ..., w_J) - \eta \left( \nabla_{\alpha, w_1, ..., w_J} \widehat{R} \left( \sum_{j \in [J]} \alpha_j h_{j,w_j} \right) + C_{\ell_1}(\alpha) \right)$;
5          $\alpha \leftarrow P_{\Delta^J}(\alpha)$;
6      **end**
7      $\lambda \leftarrow 2\lambda$
8  **end**

---

## 3 APPLYING DARC TO DEEP NETWORKS

DARC can be applied in a myriad of ways to compress deep neural networks. In all of these ways, the basic premise is to intelligently replace components of the network with cheaper components.

Consider a Neural Network (NN) with $L$ layers. For layer $\ell$, let $W_\ell$ be the parameters of the layer, and $g_\ell$ be the function mapping inputs and parameters to the output (in layers having no parameters, $W$ can be an empty token). For example, for a fully connected layer, $W$ is a matrix, the input $x$ is a vector, and $g$ is the matrix-vector multiplication function. We can write the NN as a function:

$$f(x) = g_L(W_L, g_{L-1}(W_{L-1}, \cdots g_1(W_1, x) \cdots )), \tag{2}$$

To apply DARC, we consider a set of replacement candidates $(g_{\ell,2}, W_{\ell,2}), ..., (g_{\ell,J_\ell}, W_{\ell,J_\ell})$ for each layer $\ell$ (with $g_{\ell,1}, W_{\ell,1}$ denoting the original function and weight of the layer). For each candidate $j$ in layer $\ell$, DARC takes as input an associated cost $C_{\ell,j} \geq 0$. Examples of such costs include

parameters count, FLOPs, or latency, which are usually easy to calculate or estimate experimentally. Applying DARC to neural network compression then involves four main steps:

1. **Layerwise Continuous Relaxation:** First, we replace each $g_\ell, W_\ell$ with a weighted average $\widetilde{g}_\ell(\widetilde{W}_\ell, \alpha_\ell, x) = \sum_{j=0}^{J_\ell} \alpha_{\ell,j} g_{\ell,j}(W_{\ell,j}, x)$, where $\alpha_\ell \in \Delta^{J_\ell}$. The original network is replaced by $\widetilde{f}(x) = \widetilde{g}_L(\widetilde{W}_L, \alpha_L, \cdots \widetilde{g}_1(\widetilde{W}_1, \alpha_1, x) \cdots)$.

2. **DARC Model Initialization:** Before training the DARC model, we need to initialize the $\alpha$ weights and the parameters of the compression candidates. We initialized the $\alpha$ parameters as uniform vectors $\alpha_\ell = (1/J_\ell, 1/J_\ell, ..., 1/J_\ell)$. The other option we considered was to put all weight on the original candidate ($\alpha_\ell = (1, 0, ..., 0)$), so that the initial model was equivalent to the original model being compressed. However, this makes the gradient of the loss $0$ with respect to all parameters of the compression candidate, preventing these from training. Furthermore, the non-convex regularization discourages the weights of $\alpha_\ell$ to shift towards a value that makes use of the compression candidates. As for candidate parameters, we initialized each compression candidate to mimic the original layer, which we know gives good prediction results. In some cases, this can be done analytically (e.g. via PCA for lower dimensional fully-connected layers); more generally, this can be done via SGD, training the new candidate to minimize squared loss between its outputs and those of the original layer $g_{\ell,1}(W_{\ell,1}, x)$. Since this is only for initialization, it suffices to use a small training sample and crude optimization procedure.

3. **Training the Relaxed Model:** We minimize the empirical risk, simultaneously over the mixture weights ($\alpha$s) and the candidate weights ($\widetilde{W}$s) as described in Algorithm 1.

4. **Selecting a Sub-Model:** As discussed above, for sufficiently large $\lambda$, Algorithm 1 converges to a solution with small (weighted) $\ell_0$ norm; i.e., $\alpha_\ell$ will have a small number of non-zero entries. Thus, we remove candidate $g_{\ell,j}$ (and its weight $\alpha_{\ell,j}$) from the network if $\alpha_{\ell,j} = 0$.

During optimization, we jointly optimize $\alpha$ and the model parameter on the same data, contrasting from other gradient based NAS approaches (Liu et al., 2018b) that split data into two training sets, optimize model parameters on one and $\alpha$ weights on the other. In Section 4 we analyze the Rademacher complexity of our procedure in a simple setting and show that under the condition that the original model class defined by $g_{\ell,1}$ is richer than the alternatives, optimizing all parameters jointly does not hurt generalization guarantees when compared to the original optimization objective where $J_\ell = 1$. Unlike for NAS, this condition holds naturally for model compression.

**Efficient Approximate Convolutions** Computation in most deep networks used in computer vision problems, such as image classification, image segmentation, and object detection, is dominated by convolutional layers. This has motivated several papers on efficient approximations to convolution, such as depthwise-separable convolution (Jaderberg et al., 2014; Zhang et al., 2016; Howard et al., 2017), bottleneck convolution (Sandler et al., 2018), and shifts (Wu et al., 2017).

In a standard convolution layer we have $k \times k$ filters for every input and output channel. Denoting the output channels by $Y_i$ and the input channels by $X_j$, the $i$'th output channel is defined as $Y_i = \sum_j X_j * F_{i,j}$. Here $F_{i,j}$ is the appropriate filter and $*$ is the convolution operator. Restricting the discussion to the setting where the number of output and input channels are the same, a fully-grouped convolution is a more constrained alternative in which the filter is $k \times k$ but each output is computed based on a single input channel. A depthwise-separable convolution consists of a full-grouped convolution followed by a standard $1 \times 1$ convolution. In most setting this operation requires less compute and memory resources. A shift layer is an even cheaper alternative to depthwise-separable where the $F_i$'s are fixed and have only a single non-zero element, resulting in computational complexity equivalent to a single $1 \times 1$ convolution. We use DARC to compress CNNs by considering alternatives from among the above options, for each convolution layer.

## 4 THEORETICAL RESULTS

Here, we discuss generalization power models learned by DARC. We restrict our attention to the simple case of learning an ensemble of models; as described below, the result has implications for our algorithm for training DARC. This setting actually applies not only for DARC but also for various NAS methods such as DARTS (Liu et al., 2018b) or ENAS (Pham et al., 2018). Indeed, these methods aim to choose one out of several options in each layer. While these methods differ in how this ensemble is learned, our generalization bound is independent of the learning technique.

Recall that DARC learns a convex combination of functions from $J$ classes $\mathcal{H}_1, \ldots, \mathcal{H}_j$. Here, we analyze generalizability of this process via Rademacher complexity Bartlett & Mendelson (2003):

**Definition 1** (Rademacher Complexity). Let $\mathcal{H}$ be a class of functions mapping $\mathcal{X} \to \mathbb{R}$ and let $n \in \mathbb{N}$. Denote by $X_1^n = (X_1, ..., X_n)$ $n$ IID samples from $\mathcal{X}$. Let $\sigma$ a uniform random vector in $\{-1, 1\}^n$. The Rademacher complexity of $\mathcal{H}$ is $\mathfrak{R}(\mathcal{H}) = \mathbb{E}_{\sigma, X_1^n} \left[ \sup_{h \in \mathcal{H}} \frac{1}{n} \sum_{i \in [n]} \sigma_i h(X_{n,i}) \right]$.

It is well known that the Rademacher complexity of a class $\mathcal{H}$ is equal to that of the convex hull of $\mathcal{H}$. Based on this fact, for $h = (h_1, \ldots, h_J) \in \prod_{j=1}^{J} \mathcal{H}_j, \alpha \cdot h = \sum_{j=1}^{J} \alpha_j h_j$, we have a generalization bound on the difference between true risk $R$ and empirical risk $\widehat{R}$:

**Theorem 1.** *Suppose we jointly estimate $\alpha, h_1, ..., h_J$; i.e., $\left( \widehat{\alpha}, \widehat{h}_1, ..., \widehat{h}_J \right) := \arg\min_{h_j \in \mathcal{H}_j, \alpha \in \Delta^J : C \cdot \alpha \leq B} \sum_{i=1}^{n} \mathcal{L} (\alpha \cdot h(X_i), Y_i)$. Let $\mathcal{L}(h(x), y) = 1_{\{f(x) \neq y\}}$ be 0-1 loss. Then, w.p. $\geq 1 - \delta$ (over $n$ training samples), $R\left( \widehat{\alpha} \cdot \widehat{h} \right) - \widehat{R}\left( \widehat{\alpha} \cdot \widehat{h} \right) \leq \mathfrak{R}\left( \bigcup_{j \in [J]} \mathcal{H}_j \right) + \sqrt{\frac{\log 1/\delta}{n}}$.*

Since Theorem 1 follows from standard Rademacher generalization bounds (e.g., (Bartlett & Mendelson, 2003, Theorem 5(b))), we omit its proof. According to Theorem 1, generalization error depends on a standard $\sqrt{(\log 1/\delta)/n}$ term and Rademacher complexity of the union of classes $\mathcal{H}_1, ..., \mathcal{H}_J$. If $\mathcal{H}_1, ..., \mathcal{H}_J$ are diverse, this union can be quite rich and so $\mathfrak{R}\left( \cup_{j \in [J]} \mathcal{H}_j \right)$ might be large, leading to overfitting. However, consider an example where $\mathcal{H}_1$ is the family of full convolutions, $\mathcal{H}_2$ is the family of depthwise-separable convolutions, $\mathcal{H}_3$ is the family of sparse convolutions, etc. Here, we actually have $\mathcal{H}_1 = \bigcup_{j \in [J]} \mathcal{H}_j$; thus, Rademacher complexity is simply that of the original model (i.e., with $J = 1$). Formally:

**Corollary 1.** *Suppose that every sub-model is contained in $\mathcal{H}_1$; i.e., $\mathcal{H}_2, ..., \mathcal{H}_J \subseteq \mathcal{H}_1$. Then, the Rademacher complexity of DARC is at most that of the original model: $\mathfrak{R}(\mathcal{H}) \leq \mathfrak{R}(\mathcal{H}_1)$.*

Even if $\mathcal{H}_1 \subsetneq \bigcup_{j \in [J]} \mathcal{H}_j$, in the setting of model compression the alternative families $\mathcal{H}_j, j > 1$ are cheaper replacements for $\mathcal{H}_1$ suggesting that $\mathfrak{R}(\mathcal{H}_1) \approx \mathfrak{R}(\bigcup_{j \in [J]} \mathcal{H}_j)$. This observation motivates our learning framework – it shows us that there is no need to split the training set, train the model parameters on one split and the control parameters on the other, as in many NAS papers (Cai et al., 2018; Liu et al., 2018b).

We note that does not motivate a change in the learning framework of NAS. A key difference between NAS and DARC is that candidate models $\mathcal{H}_1, ..., \mathcal{H}_J$ in NAS are intentionally diverse; their union is much richer than any individual. This translates to large $\mathfrak{R}\left( \bigcup_{j \in [J]} \mathcal{H}_j \right)$, motivating a need to avoid jointly optimizing $\alpha$ and model parameters on a single training set. When keeping a validation set aside for training $\alpha$, given the limited number of update steps typical in NAS papers, generalization error may be closer to the setting of fixed $\alpha$, wherein Rademacher complexity is bounded by $\sum_j \alpha_j \mathfrak{R}(\mathcal{H}_j)$ (Cortes et al., 2014), potentially much smaller than in Theorem 1.

## 5 EXPERIMENTAL RESULTS

We applied DARC to a number of deep networks from the GluonCV Model Zoo (Mod, 2018) for image classification on the CIFAR-10 and ImageNet (Russakovsky et al., 2015) datasets; this section presents quantitative and qualitative results. We report three performance metrics (model size, single-sample throughput, and batch throughput), and we specifically considered using DARC to minimize two of these (model size and batch throughput). Below, "DARC(S)" denotes DARC with a model (S)ize penalty, and "DARC(T)" denotes DARC with batch (T)hroughput penalty.

**Choice of Compression Candidates** For each convolutional layer, besides the original (full) convolution, we considered 3 compression candidates: a (fully-grouped) depthwise-separable convolution with $3 \times 3$ kernels (abbreviated henceforth as "3x3DS"), a full convolution with $1 \times 1$ kernels ("1x1FC"), and a 2-layer candidate consisting of a 3x3DS layer followed immediately by a 1x1FC layer ("3x3+1x1"). In ResNet50, which was already implemented using bottleneck convolutions (He et al., 2016) consisting of a sequence of $1 \times 1$, $3 \times 3$, and $1 \times 1$ convolutions, the entire bottleneck

sub-network was treated as a single component (i.e., all 3 convolutions were replaced with a single block from the above mentioned alternatives). We intentionally limited the choice of alternatives to maintain a simple system that can enjoy high throughput without special implementation. Our precise choice of alternatives is motivated by them already being an established component for deep networks, proven to work in some settings even without architecture search.

**DARC Training Details** To maximize fairness when comparing with models in the GluonCV Model Zoo (Mod, 2018), most aspects of training DARC were based on the training scripts provided publicly by the Model Zoo[1]. Due to space constraints, these implementation details are decribed in Appendix A; a few specific differences from these scripts are described below.

*Student-Teacher Initialization:* As described in Section 3 we initialized compression candidates to mimic original layers using student-teacher training. While this training had to be performed separately for each compression candidate in each layer of the original model, since each compression candidate has few parameters, each candidate's training converged quite quickly. Thus, in CIFAR-10 experiments, we simply ran 1 epoch of the entire training dataset; in ImageNet experiments, we ran only 1000 batches. A relatively large step size of 0.1 was used, since teacher-student initialization is only for initialization and fine-tuning can be performed during model-selection.

*Main Training Phase:* As noted in Algorithm 1, training occurred in blocks of epochs (20 epochs/block for CIFAR-10, 10 for ImageNet), with the compression penalty $\lambda$ increased after each block, to obtain a spectrum of compressed models. For each dataset, penalty type, and model size, the initial value of $\lambda$ was selected to roughly balance the orders of magnitude of the empirical loss and the regularization term at the beginning of training. After each block, we: 1) remove candidates $j$ with $\alpha_j = 0$l, 2) save (for evaluation later) a copy of the DARC model, in which, in any layer with multiple non-zero $\alpha$ entries, all but the most expensive remaining candidate are removed, and 3) decrease learning rate $\eta$ and increase compression penalty $\lambda$ (each by a factor of 2).

This "iterative compression" was repeated until only one compression candidate per layer remained in the DARC model. Finally, each saved model was fine-tuned for 20 epochs using only prediction loss (i.e., with $\lambda = 0$). This procedure enabled us to obtain a sequence of compressed models at progressively increasing compression levels. Moreover, this "warm-starting" improved compression speed since we only perform a total of 30 epochs per $\lambda$ value, rather than the $> 100$ epochs needed for convergence at high levels of compression.

## 5.1 CIFAR-10 RESULTS

By all metrics, DARC gave the best results when applied to very wide models such as the WideResNet series (specifically, the WideResNet16_10, WideResNet28_10, and WideResNet40_8 models (Mod, 2018)). Moreover, unlike results on other ResNets, results on WideResNets were relatively similar for both DARC(S) and DARC(T); both versions of DARC selected the 3x3+1x1 candidate for every layer. The reason for this is that very wide convolutions in WideResNet models can be replaced by depthwise separable convolutions with essentially no loss in accuracy, and improvement in throughput for both batches and for single samples ($1.4$-$1.6\times$ for each) and memory footprint ($4$-$6\times$). In the case of WideResNet16_10, DARC produces a model with latency (single sample throughput) comparable to on of the fastest CIFAR-10 model (ResNet20_v1; see Figure 1), while having accuracy within $0.8\%$ of the best model (ResNeXt29_16x64d), $3.2\%$ accuracy points above the performance of ResNet20_v1. For complete CIFAR-10 results see Appendix tables 3-4.

## 5.2 IMAGENET RESULTS

For ImageNet we compressed several ResNet models. To present the size compression results we provide Table 2 comparing accuracy change as a function of parameter reduction. We compared to previous published results compressing ResNet50 on ImageNet. To our knowledge these are the state-of-the-art results among those compressing ResNet50 on ImageNet. For compression as aggressive as $3\times$, we incur an accuracy drop of $0.86\%$ while the baseline suffers a drop of $3.26\%$. Table 1 gives throughputs of our compressed ResNet34 and ResNet50 models and throughputs of state-of-the-art competing models, on identical hardware; our compressed models outperform these competing models, due to our initialization based on pre-trained ResNet34 and ResNet50 models.

---

[1]`train_imagenet.py`, `train_mixup_cifar10.py`

While further details, including other compressed versions, are available in the Appendix (Figures 2 and 3 and Table 5), Table 2 compares prediction performance of DARC(S) with that of state-of-the-art network pruning techniques applied to ResNet50 on ImageNet, at various compression levels. In the light ($1.5\times$) compression regime, the result of (Zhuang et al., 2018) outperforms ours. The work of (Zhuang et al., 2018) complements ours, in that their novelty is in warm-starting the compressed alternative not only to mimic the original, but also to be informative w.r.t. the label. Since this work does not have an architecture search component, we suggest that future work combine this clever warm-start with an architecture search component such as DARC. Once the compression becomes more aggressive, DARC outperforms the baseline, likely since in that regime (smaller network with more training epochs), a good architecture is more important than a good warm-start.

| Model | Accuracy (Top-1) | Throughput ($224 \times 224$px/s) |
|---|---|---|
| ResNet34 (Mod, 2018) | 74.4 | 205 |
| ResNet50 (Mod, 2018) | 79.1 | 148 |
| ProxylessNAS* (Cai et al., 2018) | 75.1 | 196 |
| MobileNetV2* (Sandler et al., 2018) | 72.0 | 164 |
| MnasNet* (Tan et al., 2019) | 74.0 | 164 |
| DARTS* (Liu et al., 2018b) | 73.1 | — |
| DARC-ResNet34 (ours) | 73.9 | 234 |
| DARC-ResNet50 (ours) | 76.8 | 208 |

Table 1: Comparison of ResNet34/ResNet50 compressed by DARC(T) with models from state-of-the-art competing methods (as reported by Cai et al. (2018) in an identical runtime environment).

## 5.3 DISCUSSION OF COMPRESSED ARCHITECTURES

In both ResNets and WideResNets, model size tends to be dominated by a small number of the largest layers in the model, and is relatively insensitive to the depth of the network. Thus, significant compression can be achieved by replacing these large layers with 3x3+1x1 candidates, which offer compression of nearly $9\times$ (for $3 \times 3$ convolutions). Since the sizes (number of convolutional kernels) of ResNet layers increases from bottom to top (i.e., from input to output) DARC(S) tends to first replace the top-most layers (i.e., layers closest to the output) of the network with 3x3+1x1 candidates, proceeding towards the bottom of the network as the compression parameter $\lambda$ increases.

| ResNet-50 | Compression | Top1/Top5 |
|---|---|---|
| Disc | $1.51\times$ | $+\mathbf{0.39}/+\mathbf{0.14}$ |
| DARC(S) | $1.63\times$ | $-0.51/-0.16$ |
| ThiNet | $2.06\times$ | $-1.87/-1.12$ |
| Disc | $2.06\times$ | $-1.06/-0.61$ |
| DARC(S) | $2.05\times$ | $-\mathbf{0.63}/-\mathbf{0.19}$ |
| Disc | $2.94\times$ | $-3.26/-1.80$ |
| DARC(S) | $3.01\times$ | $-\mathbf{0.86}/-\mathbf{0.28}$ |
| DARC(S) | $3.57\times$ | $-0.99/-0.35$ |
| DARC(S) | $6.14\times$ | $-6.09/-4.85$ |

Table 2: Comparison of DARC and state-of-the-art pruning methods, Disc (Zhuang et al., 2018) and ThiNet (Luo et al., 2017), when compressing ResNet-50 trained ImageNet for ResNet-50. (S)ize denotes models optimized by DARC to minimize model size. Top1/Top5 accuracy of pre-trained model were 79.15%/94.58% respectively.

In contrast, model latency is relatively uniformly distributed throughout the layers of the network – the time taken to compute each convolutional layer scales only weakly with the number, size, and grouping of filters. Thus, in ResNets (but not in WideResNets) 3x3+1x1 candidates, which replace 1 layer with 2 smaller layers, tend to offer little or no benefit in throughput; of the compression candidates we considered, only the smallest (3x3DS) candidates offer significant acceleration (typically of about $2\times$) over full convolutions. As a result, the acceleration offered by DARC(T) scales primarily with the number of layers that can be replaced by 3x3DS layers. Moreover, the replaced layers tend to be scattered throughout the network (rather than clustered near the output of the network). As noted earlier, in WideResNets, each layer is so large that the 3x3+1x1 candidate *is* much faster than full convolution, and DARC(T) selects this candidate for all layers.

Overall, we see that the optimal compression strategy depends on whether one is optimizing for size or for speed. This implies that some sort of intelligent replacement of components, as in DARC, is needed to obtain reliable performance improvements (as opposed to, say random baselines that have been shown to perform well in pruning). This parallels recent work (Cai et al., 2018) showing (for NAS) that optimizing for speed on different hardware (GPU, CPU, or mobile) leads to different

models. While we focused on GPU throughput, we note that the speedup of depthwise-separable convolution over full convolution is typically larger on CPU and embedded devices than on GPU devices. Thus, DARC should also produce efficient architectures for these alternative hardware.

## 6 RELATED WORK

Work on compressing deep networks has abounded in recent years, with diverse approaches including pruning (LeCun et al., 1990; Polyak & Wolf, 2015; Li et al., 2016a; He et al., 2017; Luo & Wu, 2017; Luo et al., 2017; Zhuang et al., 2018), low-rank factorization (Jaderberg et al., 2014; Zhang et al., 2016; Howard et al., 2017), fast approximate convolutions (Bagherinezhad et al., 2017; Wu et al., 2017), knowledge distillation (Hinton et al., 2015; Romero et al., 2014), and quantization (Gong et al., 2014; Han et al., 2015; Zhou et al., 2017; Lin et al., 2016); (Cheng et al., 2018) survey common approaches. In contrast, DARC searches a richer space of alternative models and has the ability to replace multiple components by a single one (e.g. replacing a bottleneck sub-network of 3 convolutions with a single convolution). Indeed, though beyond the scope of this paper, DARC can incorporate or complement many of these methods. Our initialization of compression candidates by mimicking the original layer is also reminiscent of knowledge distillation.

Another closely-related line of papers concerns Neural Architecture Search (NAS) (Pham et al., 2018; Kandasamy et al., 2018; Liu et al., 2018b; Gordon et al., 2018; Cai et al., 2018). The most relevant papers are (Liu et al., 2018b), (Gordon et al., 2018), and (Cai et al., 2018), who all use a sparse linear component weighting scheme similar to DARC. (Liu et al., 2018b) focused on pure architecture search, in which the goal is simply to find an architecture maximizing prediction performance, without consideration of inference-time efficiency. (Gordon et al., 2018) do not aim to replace layers with general alternatives but rather discover parameters in a data-driven way; their experiments are restricted to results in channel pruning. Recently, (Cai et al., 2018) performed NAS with a latency regularization term similar to ours.

Our methods differ from these NAS papers in two main ways (summarized in Table 3). First, our architecture search is guided by an established base model that was already tested and proven useful. This distinction allows a simpler and more efficient learning scheme (motivated in Section 4) that avoids iterating between training sets, to optimize model and $\alpha$ parameters. Furthermore, starting with a pre-trained model allows us not only to reach an effective architecture, but to warm start the weight parameters. As evidence of the advantage of starting with a pretrained model, our compressed model ResNet50(T) on ImageNet has Top1 accuracy $\geq 3\%$ more than models obtained by these NAS papers; thus, it seems that starting architecture search with a highly accurate base model can improve the efficiency/accuracy trade-off of the learned model. Another distinction from gradient-based NAS results is our use of sparsity-inducing regularization. Previous methods make the choice between the candidates via a softmax layer; this restricts the output to be a convex combination of inputs without optimizing sparsity. Since these methods also aim to find a sparse combination, they might benefit from a non-convex regularization term as in DARC.

Table 3: Comparison of DARC with NAS methods. "SGD" denotes stochastic gradient descent. "RL" denotes reinforcement learning.

| Method | Optimizer | Efficient architecture | Resists Overfitting | Exploits pre-training | Iterative compression |
|---|---|---|---|---|---|
| DARTS (Liu et al., 2018b) | SGD | ✗ | ✗ | ✗ | ✗ |
| ENAS (Pham et al., 2018) | RL | ✗ | ✗ | ✗ | ✗ |
| ProxylessNAS (Cai et al., 2018) | SGD/RL | ✓ | ✓ | ✗ | ✗ |
| MNasNet (Tan et al., 2019) | SGD | ✓ | ✓ | ✗ | ✗ |
| FBNet (Wu et al., 2019) | SGD | ✓ | ✓ | ✗ | ✗ |
| DARC (proposed method) | SGD | ✓ | ✓ | ✓ | ✓ |

## 7 CONCLUSIONS AND FUTURE WORK

We have shown that even a simple DARC implementation, with only depthwise-separable approximations as compression candidates, can be used to compress large state-of-the-art deep networks, improving inference speed and memory footprint. Intelligently making only some layers of the net-

work depthwise-separable results in compressed models with much better predictive performance than simply making *all* convolutions depthwise-separable, as in (Howard et al., 2017).

While depthwise-separable convolutions are easily implemented in existing deep learning packages and already offer substantial compression, future work may benefit from more sophisticated approximate convolutions, with efficient implementations. For example, shift operations (Wu et al., 2017) are promising, as they require no stored parameters and replace the slow multiplications in convolution with fast indexing. Another venue worth pursuing is to compress a model into a shallower version. Although there are a few ways this could be attempted, such as an Identity candidate or replacing entire blocks of layers, it is unclear which technique would work best. Finally, we hope to apply DARC to models other than CNNs (e.g., models with recurrent cells or transformers).

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

# A  DARC IMPLEMENTATION DETAILS

In this section, we provide further details about the implementation of DARC used in our experiments.

**Environment Details**  We implemented DARC in Apache MXNet 1.3.1 using Python 3.6 and CUDA 9.0. Experiments were run on AWS EC2 p3.8xlarge and p3.16xlarge machines, which respectively features 4 and 8 NVIDIA Tesla-V100 GPUs. CIFAR-10 models were each trained with 1 GPU. Smaller ImageNet models (ResNet18 and ResNet34) were trained using 4 GPUs, while the larger ResNet50 was trained using 8 GPUs.

**Training Details**  Following the original script used to train models in the GluonCV Model Zoo, we utilized mixup training (Zhang et al., 2017), and optimized cross-entropy loss with Nesterov accelerated stochastic gradient descent (NAG) with (default) momentum parameter $0.9$. As noted in the main paper, for student-teacher initialization, we used a relatively large learning rate $\eta = 0.1$. Thereafter, for model-selection, we began with an initial learning rate of $\eta = 0.01$, which was then halved after each traning block.

To minimize training time, training batch sizes were selected to be as large as possible without exceeding GPU memory during training. This resulted in batch sizes (per training GPU) of $256$ for ResNets on CIFAR-10, $128$ for WideResNets on CIFAR-10, $64$ for ResNet18 and ResNet34 on ImageNet, and $32$ for ResNet50 on ImageNet.

For each dataset, penalty type, and model size, the initial value of $\lambda$ was selected to roughly balance the orders of magnitude of the empirical loss and the regularization term at the beginning of training. For CIFAR-10 experiments with size penalization, the initial value of the $\lambda$ compression penalty was set to $\lambda = 10^{-5} \times L$, where $L$ is the number of layers to which DARC was applied (i.e., the number of full convolutions in the original model). For CIFAR-10 experiments with latency penalization, we used $\lambda = 10^{4} \times L$. For ImageNet experiments, we used $\lambda = 10^{-8} \times L$ with size penalization and $L = 10^{4} \times L$ for latency penalization.

## A.1  MEASURES OF MODEL PERFORMANCE

**Computational Performance**  As an estimate of model size, we report the size (on disk) of the parameter file created by MXNet when saving the model; this correlates well with both the number of parameters in the model and the footprint of the model in RAM or GPU memory. Since throughputs are inherently noisy, we report average inference times over $1000$ batches. Though multiple GPUs were used for training DARC, all inference times were computed using a single Tesla V100 GPU. We used batch size $1$ to estimate single-sample throughput and batch size $256$ to estimate batch throughput. The cost of each compression candidate (i.e., number of parameters for DARC(S) or latency for DARC(T)) was calculated or estimated based on the student model trained during initialization.

**Prediction Performance**  On CIFAR-10, we used standard ("Top1") prediction accuracy. On ImageNet, we additionally used ("Top5") accuracy, the fraction of test images for which the correct label is among the five labels considered most probable by the model. We note that these are the standard performance used for these datasets (Krizhevsky & Hinton, 2010; Krizhevsky et al., 2012).

# B SUPPLEMENTARY RESULTS

This section provides detailed numerical results of our experiments:

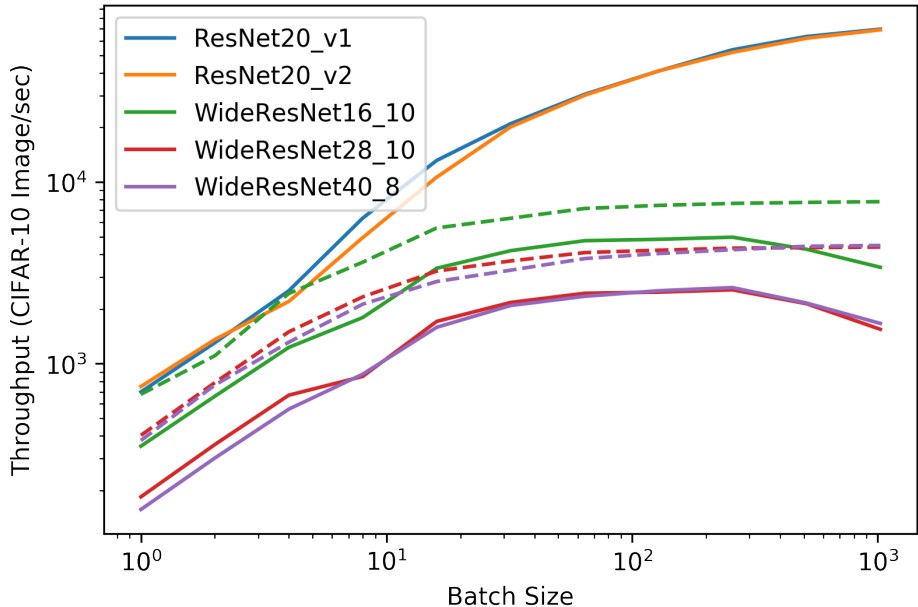

Figure 1: Throughputs of original and compressed WideResNet models, and of original ResNet20 models, at various batch sizes.

| Model | Top-1 | Model Size | Throughput (1 Im/batch) | Throughput (256 Im/batch) |
|---|---|---|---|---|
| ResNet20_v1 (O) | 92.9 | 1.05 | 709.62 | 53020.76 |
| ResNet20_v1 (S) | 92.1 | 0.90 | 682.85 | 53078.90 |
| ResNet20_v1 (S) | 92.0 | 0.78 | 706.91 | 52822.23 |
| ResNet20_v1 (S) | 90.2 | 0.60 | 699.80 | 53206.18 |
| ResNet20_v1 (S) | 89.0 | 0.26 | 723.89 | 53058.42 |
| ResNet20_v2 (O) | 92.7 | 1.05 | 792.03 | 52889.67 |
| ResNet20_v2 (S) | 92.1 | 0.85 | 665.98 | 53619.09 |
| ResNet20_v2 (S) | 91.4 | 0.67 | 671.05 | 54148.44 |
| ResNet20_v2 (S) | 91.0 | 0.48 | 700.55 | 54144.82 |
| ResNet20_v2 (S) | 90.3 | 0.26 | 663.76 | 56124.30 |
| ResNet20_v2 (S) | 89.1 | 0.25 | 739.73 | 55545.69 |
| ResNet56_v1 (O) | 94.2 | 3.31 | 345.89 | 18930.70 |
| ResNet56_v1 (S) | 93.4 | 1.60 | 369.71 | 19974.62 |
| ResNet56_v1 (S) | 93.3 | 1.34 | 348.71 | 20330.08 |
| ResNet56_v1 (S) | 92.9 | 1.18 | 347.43 | 20711.30 |
| ResNet56_v1 (S) | 92.8 | 0.60 | 354.43 | 21155.00 |
| ResNet56_v1 (S) | 92.7 | 0.59 | 343.84 | 21648.25 |
| ResNet56_v2 (O) | 94.6 | 3.30 | 395.35 | 19266.32 |
| ResNet56_v2 (S) | 93.4 | 1.12 | 382.69 | 20138.11 |
| ResNet56_v2 (S) | 93.2 | 1.12 | 360.47 | 20587.73 |
| ResNet56_v2 (S) | 93.1 | 0.93 | 353.93 | 20874.39 |
| ResNet56_v2 (S) | 92.9 | 0.71 | 355.93 | 20993.20 |
| ResNet56_v2 (S) | 92.7 | 0.63 | 335.27 | 21578.89 |
| ResNet56_v2 (S) | 92.5 | 0.59 | 330.72 | 22315.29 |
| ResNet110_v1 (O) | 95.2 | 6.68 | 239.89 | 9704.98 |
| ResNet110_v1 (S) | 94.1 | 2.84 | 213.33 | 10391.16 |
| ResNet110_v1 (S) | 94.1 | 2.38 | 198.18 | 10704.93 |
| ResNet110_v1 (S) | 94.1 | 1.10 | 209.27 | 10858.91 |
| ResNet110_v1 (S) | 93.9 | 1.10 | 203.75 | 11084.18 |
| ResNet110_v1 (S) | 93.7 | 1.09 | 191.06 | 11245.70 |
| ResNet110_v2 (O) | 95.5 | 6.68 | 220.21 | 9942.27 |
| ResNet110_v2 (S) | 94.3 | 3.13 | 210.58 | 10393.19 |
| ResNet110_v2 (S) | 94.3 | 2.08 | 209.11 | 10523.53 |
| ResNet110_v2 (S) | 94.2 | 1.97 | 199.03 | 10568.06 |
| ResNet110_v2 (S) | 93.9 | 1.21 | 214.44 | 10740.52 |
| ResNet110_v2 (S) | 93.7 | 1.14 | 219.05 | 11029.72 |
| ResNet110_v2 (S) | 93.7 | 1.09 | 209.54 | 11320.08 |
| WideResNet16_10 (O) | 96.7 | 65.34 | 345.19 | 4914.29 |
| WideResNet16_10 (S) | 96.6 | 17.11 | 608.58 | 6932.60 |
| WideResNet16_10 (S) | 96.5 | 16.23 | 631.89 | 7546.51 |
| WideResNet28_10 (O) | 97.1 | 139.24 | 186.63 | 2525.49 |
| WideResNet28_10 (S) | 97.2 | 26.42 | 423.48 | 3918.92 |
| WideResNet28_10 (S) | 97.1 | 24.67 | 425.66 | 4311.35 |
| WideResNet40_8 (O) | 97.3 | 136.47 | 156.06 | 2617.89 |
| WideResNet40_8 (S) | 97.3 | 21.31 | 358.66 | 4258.24 |

Table 4: Results of applying size-penalized DARC to CIFAR-10 models. (O)riginal Denotes an original model taken from the GluonCV Model Zoo (Mod, 2018). (S)ize denotes a model optimized by DARC to minimize model size. (T)hroughput denotes a model optimized by DARC to maximize model throughput. Model Size is provided in MB on disk. Throughput numbers are provided in 32px ×32px images/second.

| Model | Top-1 | Model Size | Throughput (1 Im/batch) | Throughput (256 Im/batch) |
|---|---|---|---|---|
| ResNet20_v1 (O) | 92.9 | 1.05 | 684.32 | 52239.12 |
| ResNet20_v1 (T) | 91.4 | 1.03 | 749.98 | 34743.19 |
| ResNet20_v1 (T) | 90.6 | 0.93 | 1144.10 | 48822.26 |
| ResNet20_v1 (T) | 88.3 | 0.92 | 1233.44 | 69868.65 |
| ResNet20_v2 (O) | 92.7 | 1.05 | 737.07 | 52336.75 |
| ResNet20_v2 (T) | 91.9 | 1.01 | 691.04 | 60078.59 |
| ResNet20_v2 (T) | 90.7 | 0.83 | 1178.61 | 59269.61 |
| ResNet20_v2 (T) | 90.2 | 0.83 | 1275.66 | 70498.54 |
| ResNet56_v1 (O) | 94.2 | 3.31 | 377.22 | 18966.31 |
| ResNet56_v1 (T) | 93.5 | 3.05 | 389.21 | 19175.07 |
| ResNet56_v1 (T) | 92.9 | 2.97 | 398.94 | 24985.82 |
| ResNet56_v1 (T) | 92.5 | 2.94 | 399.47 | 25682.29 |
| ResNet56_v2 (O) | 94.6 | 3.31 | 389.76 | 18966.31 |
| ResNet56_v2 (T) | 94.4 | 3.16 | 398.09 | 20293.55 |
| ResNet56_v2 (T) | 94.3 | 3.16 | 409.56 | 20382.47 |
| ResNet56_v2 (T) | 94.3 | 3.15 | 409.78 | 21752.16 |
| ResNet110_v1 (O) | 95.2 | 6.68 | 213.59 | 9615.24 |
| ResNet110_v1 (T) | 94.5 | 5.50 | 222.80 | 10161.07 |
| ResNet110_v1 (T) | 93.1 | 5.48 | 229.85 | 12565.72 |
| ResNet110_v1 (T) | 90.3 | 5.47 | 237.86 | 13032.99 |
| ResNet110_v2 (O) | 95.5 | 6.68 | 174.33 | 9160.58 |
| ResNet110_v2 (T) | 94.9 | 5.35 | 225.93 | 10892.03 |
| ResNet110_v2 (T) | 94.5 | 5.20 | 226.46 | 11331.41 |
| ResNet110_v2 (T) | 94.3 | 5.19 | 233.13 | 11472.62 |
| WideResNet16_10 (O) | 96.7 | 65.34 | 349.98 | 5019.10 |
| WideResNet16_10 (T) | 96.1 | 16.23 | 601.01 | 7699.13 |
| WideResNet28_10 (O) | 97.2 | 139.24 | 158.52 | 2553.99 |
| WideResNet28_10 (T) | 96.8 | 24.67 | 455.45 | 4339.26 |
| WideResNet40_8 (O) | 97.3 | 136.47 | 155.67 | 2643.25 |
| WideResNet40_8 (T) | 96.4 | 21.31 | 395.71 | 4269.20 |

Table 5: Results of applying speed-penalized DARC to CIFAR-10 models. (O)riginal Denotes an original model taken from the GluonCV Model Zoo (Mod, 2018). (S)ize denotes a model optimized by DARC to minimize model size. (T)hroughput denotes a model optimized by DARC to maximize model throughput. Model Size is provided in MB on disk. Throughput numbers are provided in 32px $\times$32px images/second.

| Model | Top-1 | Model Size | Throughput (1 Im/batch) | Throughput (256 Im/batch) |
|---|---|---|---|---|
| ResNet18_v1 (O) | 70.9 | 44.64 | 333.86 | 4044.52 |
| ResNet18_v1 (S) | 69.7 | 37.47 | 391.47 | 4142.90 |
| ResNet18_v1 (S) | 69.6 | 30.72 | 399.27 | 4196.04 |
| ResNet18_v1 (S) | 68.3 | 14.10 | 421.47 | 4281.12 |
| ResNet18_v1 (T) | 69.9 | 42.43 | 417.64 | 4296.34 |
| ResNet18_v1 (T) | 69.6 | 38.98 | 449.36 | 4473.29 |
| ResNet18_v1 (T) | 67.9 | 36.77 | 457.60 | 5173.91 |
| ResNet34_v1 (O) | 74.4 | 83.23 | 205.15 | 2441.79 |
| ResNet34_v1 (S) | 73.5 | 37.68 | 216.68 | 2480.66 |
| ResNet34_v1 (S) | 72.4 | 31.56 | 220.29 | 2543.27 |
| ResNet34_v1 (T) | 73.9 | 70.66 | 233.92 | 2693.20 |
| ResNet34_v1 (T) | 73.2 | 60.77 | 245.57 | 3289.19 |
| ResNet50_v1 (O) | 79.1 | 97.79 | 147.61 | 1242.42 |
| ResNet50_v1 (S) | 78.6 | 59.93 | 150.82 | 1257.26 |
| ResNet50_v1 (S) | 78.5 | 47.71 | 157.79 | 1263.03 |
| ResNet50_v1 (S) | 78.3 | 32.49 | 161.64 | 1298.97 |
| ResNet50_v1 (S) | 78.2 | 27.42 | 167.92 | 1359.69 |
| ResNet50_v1 (S) | 73.1 | 15.93 | 175.34 | 1492.43 |
| ResNet50_v1 (T) | 78.3 | 92.57 | 175.68 | 1518.25 |
| ResNet50_v1 (T) | 78.2 | 71.19 | 199.76 | 1602.78 |
| ResNet50_v1 (T) | 76.8 | 69.29 | 208.01 | 1682.92 |
| ResNet101_v1 (O) | 77.2 | 170.54 | 100.25 | 724.79 |
| ResNet152_v1 (O) | 78.1 | 230.49 | 69.27 | 500.19 |
| MobileNet1.0 (O) | 69.5 | 16.24 | 597.59 | 4393.48 |
| MobileNet0.75 (O) | 66.2 | 9.94 | 627.24 | 6161.25 |
| MobileNet0.5 (O) | 61.2 | 5.13 | 662.31 | 9805.38 |
| MobileNetV2_1.0 (O) | 70.2 | 13.53 | 369.75 | 3615.16 |
| MobileNetV2_0.75 (O) | 68.1 | 10.15 | 376.56 | 4739.72 |
| MobileNetV2_0.5 (O) | 64.0 | 7.59 | 398.03 | 7065.31 |
| VGG11 (O) | 66.9 | 506.83 | 318.53 | 912.44 |
| VGG13 (O) | 68.0 | 507.54 | 262.04 | 613.30 |
| VGG16 (O) | 71.5 | 527.79 | 205.29 | 458.85 |
| VGG19 (O) | 72.9 | 548.05 | 173.37 | 363.46 |
| DenseNet121 (O) | 74.0 | 30.80 | 119.88 | 1104.38 |
| DenseNet161 (O) | 76.9 | 110.31 | 63.33 | 508.16 |
| DenseNet169 (O) | 75.5 | 54.65 | 81.86 | 881.48 |
| DenseNet201 (O) | 76.6 | 77.30 | 62.83 | 673.12 |

Table 6: Results of applying DARC to ImageNet models. (O)riginal Denotes an original model taken from the GluonCV Model Zoo (Mod, 2018). (S)ize denotes a model optimized by DARC to minimize model size. (T)hroughput denotes a model optimized by DARC to maximize model throughput. Model Size is provided in MB on disk. Throughput numbers are provided in 224px ×224px images classified/second.

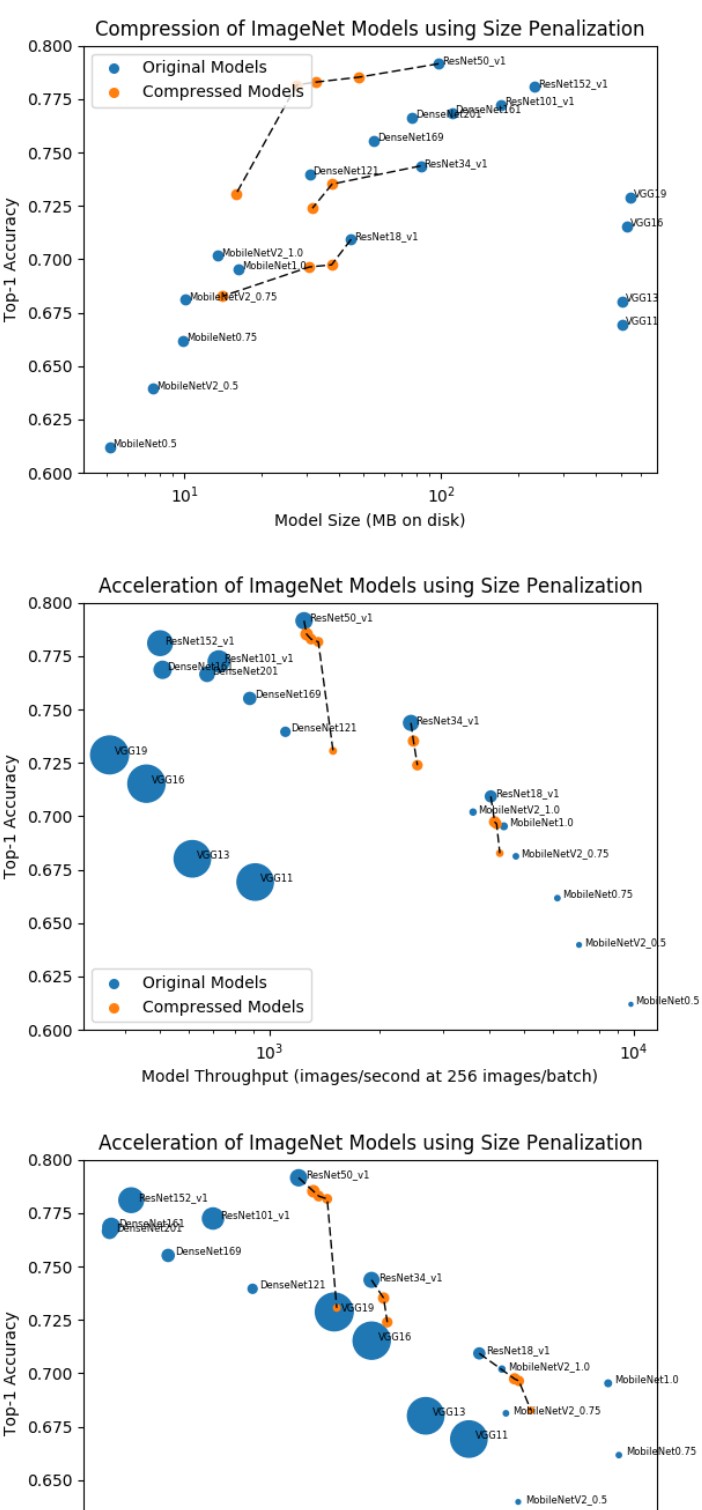

Figure 2: Compression of ImageNet models, in terms of size, throughput, and latency, when size is used as the computational penalty in DARC.

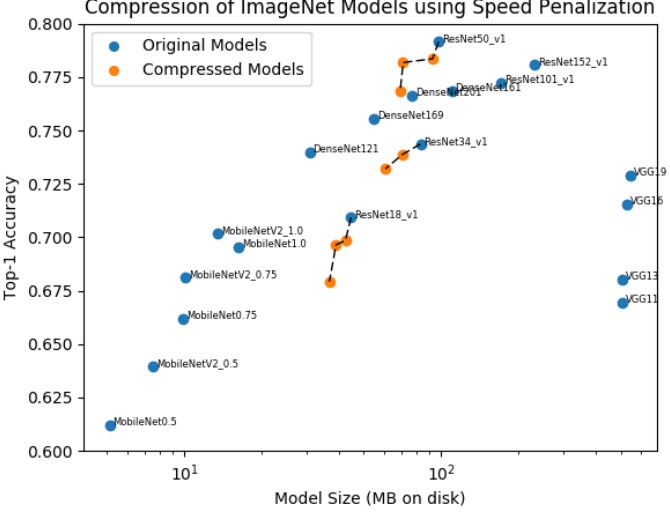

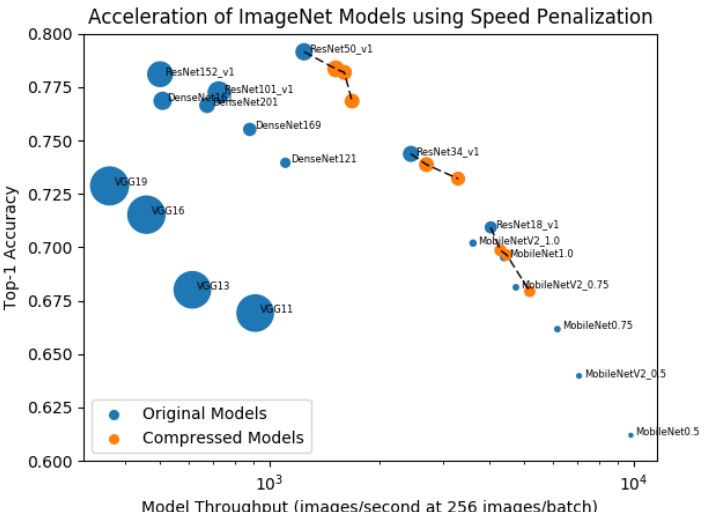

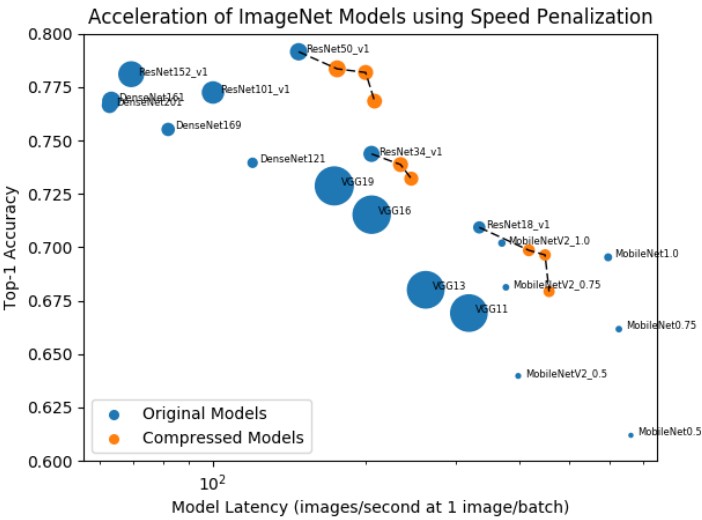

Figure 3: Compression of ImageNet models, in terms of size, throughput, and latency, when empirical latency is used as the computational penalty in DARC.

