# OpenReview forum: "Differentiable Architecture Compression"
_ICLR.cc/2020/Conference — Reject_

### Official Review · AnonReviewer1 · 2019-10-23
**Official Blind Review #1**

**Rating:** 3

**Review:**

The authors proposed a new gradient-based architecture search method that tries to find more efficient alternatives starting from the pre-trained model. The approach is similar to DARTS (Liu et al., 2019) with a budget constraint, such as size and throughput. One major difference is to modify the update of architectural parameters, i.e., mixing weights of all candidate operations, to induce the sparsity rather than to keep the weighted sum of all possible operations. Another difference is that it starts from a well-defined architecture with pre-trained weights. It is simple to apply, but hard to tell. The recognized strengths and concerns are as follows.

< Strengths >
1. The proposed update method to induce the sparsity of the architecture during the search seems applicable to all gradient-based search methods, and more important for the budget-constrained search minimizing the discrepancy between the ensemble architectures during the search and the final architecture derived after the search.
2. The setup to initialize with the well-trained model seems practically useful rather than starting the search with a random initial model.
3. The proposed methods seem easy to apply.

< Concerns & Questions >
1. The algorithm does not seem efficient because it continues training iteratively while increasing the strength (\lambda) of the L1 regularizer by the estimated cost until the budget constraint is met.
2. No details to calculate the cost C_l0(\alpha) for each architecture candidate (e.g., the throughput cost of the architecture with a 3x3 DS operation in layer 1).
3. No number of operations (e.g., # FLOPs, # MACs) is reported. Since the throughput is strongly dependent on the underlying hardware, the number of operations also needs to be shown as a more general estimation of the model inference latency in various hardware devices.
4. The models under comparison are out-dated. It needs to be compared with the latest models designed with computational efficiency in mind, e.g., EfficientNet (Tan et al. 2019a), MixNet (Tan et al. 2019b), MobileNet V3 (Howard et al., 2019).

- Liu et al., DARTS: differentiable architecture search, ICLR 2019
- Tan et al., EfficientNet: Rethinking Model Scaling for Convolutional Neural Networks, ICML 2019a
- Tan et al., MixConv: Mixed Depthwise Convolutional Kernels, arxiv:1907.09595, 2019b
- Howard et al., Searching for MobileNetV3, arxiv:1905.02244, 2019

**Experience Assessment:**

I have read many papers in this area.

**Review Assessment: Checking Correctness Of Derivations And Theory:**

I assessed the sensibility of the derivations and theory.

**Review Assessment: Checking Correctness Of Experiments:**

I carefully checked the experiments.

**Review Assessment: Thoroughness In Paper Reading:**

I read the paper thoroughly.

---

> ### Author Response · Authors · 2019-11-15
> **Author Response to Official Blind Review #1**
>
> Reviewer #1:
> Thank you for your detailed review and for raising several valid concerns and questions. We believe we are able to address each of your concerns, as described below:
>
> 1. The algorithm does not seem efficient because it continues training iteratively... until the budget constraint is met.
>
> Response: We feel that the additional time needed to train a DARC model is not long enough to significantly impact the practical usability of DARC. Since lambda increases exponentially between training blocks, the number of blocks needed to attain the target budget is small. As an example, in our experiments, training DARC models never took more than 200 total epochs to converge (10 per lambda value for ImageNet models). Considering the original models from the GluonCV model zoo were trained for 120 epochs, this is not a dramatic increase in training time.
>
> 2. No details to calculate the cost C_l0(\alpha) for each architecture candidate (e.g., the throughput cost of the architecture with a 3x3 DS operation in layer 1).
>
> Response: When compressing the size of networks ("DARC(S)"), C_j was the number of float parameters in the jth candidate, which scales close to linearly with the size of the candidate in memory or on disk. When accelerating throughput of networks ("DARC(T)"), C_j was the latency a forward pass through the jth candidate. This was estimated by the mean (over 1000 trials) time needed to pass an input through the 1-layer "student" network that was trained during model initialization. We described this in Appendix Section A.1, under "Computational Performance", but we agree that this is a bit hard to find and will clarify this in Section 5 of the paper.
>
> 3. No number of operations (e.g., # FLOPs, # MACs) is reported. Since the throughput is strongly dependent on the underlying hardware, the number of operations also needs to be shown as a more general estimation of the model inference latency in various hardware devices.
>
> Response: Thank you for this suggestion. As noted in the previous point, in this paper, we focused on hardware-specific optimization, with costs based on empirical latencies on the target hardware. Reporting FLOPS in this case seemed less appropriate, since they were not the objective of optimization. However, we can also train DARC models that optimize FLOPS and report numbers of FLOPS, as a hardware-independent metric.
>
> 4. The models under comparison are out-dated. It needs to be compared with the latest models designed with computational efficiency in mind, e.g., EfficientNet (Tan et al. 2019a), MixNet (Tan et al. 2019b), MobileNet V3 (Howard et al., 2019).
>
> Response: Thank you for pointing out these recent papers. We will add discussion of these models to the paper, and, where possible, we will add further points of comparison.
> A few general notes: Both MixNet and MobileNet are only published as Arxiv papers and were prepared in parallel to our paper. We will write a detailed comparison still and mention that the focus of these papers is in finding the best alternative components. Our paper does not have much effort there, rather focuses on the method by which the components can be found. In that sense we feel it is complementary to these papers and a future work might improve these models using our techniques.
> In EfficientNet although the objective is similar to the methods are very different. In some sense it is the opposite of our paper since while we are providing methods to reduce the network size while having a minimal effect on accuracy, they are providing ways to increase it in order to improve accuracy. When viewed this way it is evident that our techniques are complementary.

---

### Official Review · AnonReviewer2 · 2019-10-23
**Official Blind Review #2**

**Rating:** 6

**Review:**

This paper proposes a new architecture search method called "DARC" that utilizes a differentiable objective function. Since a naive formulation of architecture search is reduced to a combinatorial optimization which is not differentiable, the optimization requires much computational cost. To overcome this difficulty, this paper proposes a L1-norm relaxation and apply such relation in a layer-wise manner. The method shares a similar spirit with NAS, but the proposed model is more like "model selection" from a fixed candidates, and thus there is a Rademacher complexity guarantee. The effectiveness of DARC is justified by thorough numerical experiments.

Although the idea is rather straight-forward, the effectiveness of the method is well supported by the thorough experiments. In particular, it works as a model compression method and shows a favorable performances compared with SOTA methods.

The pros and cons are summarized as follows.
Pros:
- The proposed method is simple and rather easy to implement.
- The numerical experiments show the proposed method gives favorable performances compared with the existing methods.

Cons:
- The idea itself is rather straight-forward.
- The theoretical analysis is instructive but its derivation does not require new techniques.
- Compared with NAS, the proposed method should prepare a set of candidates which restrict the search space. This ensures generalization but limits its flexibility.

More comments:
- I could not see how efficient the method is in terms of memory. It prepares several models in each layer, thus it requires large memories. Can it be performed on more large networks?
- The setting of C_j affects the result. How did you set C_j in the experiments?



**Experience Assessment:**

I have published one or two papers in this area.

**Review Assessment: Checking Correctness Of Derivations And Theory:**

I carefully checked the derivations and theory.

**Review Assessment: Checking Correctness Of Experiments:**

I assessed the sensibility of the experiments.

**Review Assessment: Thoroughness In Paper Reading:**

I read the paper at least twice and used my best judgement in assessing the paper.

---

> ### Author Response · Authors · 2019-11-15
> **Author Response to Official Blind Review #2**
>
> Thank you for your questions and careful review of the paper. We address each of your questions below:
>
> 1) I could not see how efficient the method is in terms of memory... Can it be performed on more large networks?
>
> Response: Although DARC indeed trains several candidates for each layer, in practice, the memory footprint of the DARC model during training is not much larger than that of the original model. This is because, by construction, the alternative candidates are typically much smaller than the original layers of the model. For very large models, or if one is considering many compression candidates, one could reduce training batch size to significantly reduce model memory footprint, at the cost of increasing training time. As an example, we were able to apply DARC to ResNet152, the largest ResNet model in the GluonCV model zoo, without needing to change the batch size from the default originally used to train the model. We will add a paragraph to Section 5 discussing the memory usage of DARC during training.
>
> 2) The setting of C_j affects the result. How did you set C_j in the experiments?
>
> Response: When compressing the size of networks ("DARC(S)"), C_j was the number of float parameters in the jth candidate, which scales close to linearly with the size of the candidate in memory. When accelerating throughput of networks ("DARC(T)"), C_j was the latency a forward pass through the jth candidate. This was estimated by the mean (over 1000 trials) time needed to pass an input through the 1-layer "student" network that was trained during model initialization. We described this in Appendix Section A.1, under "Computational Performance", but we agree that this is a bit hard to find and will clarify this in Section 5 of the paper.
>
>
> “The theoretical analysis is instructive but its derivation does not require new techniques”
>
> Response: We actually view this as a good property of the paper. There are many papers whose sole purpose is to provide theoretical justification for a heuristic performed in practice. This contributes to the scientific community by providing more guidance for ways of improving the existing techniques
>
> “Compared with NAS, the proposed method should prepare a set of candidates which restrict the search space. This ensures generalization but limits its flexibility”
>
> Response: We agree that our proposed search space is limited by construction compared to NAS, but this is a necessary artifact in an approach that takes advantage of a high quality yet resourceמו heavy model.

---

### Official Review · AnonReviewer3 · 2019-10-27
**Official Blind Review #3**

**Rating:** 6

**Review:**

This paper addresses the problem of limited resources that may generally be available at inference time (as compared to the lack of constraints during training time). The paper addresses architecture search as well as model compression by simultaneously optimizing all layers/weights/submodules of the network. The approach is motivated well in terms of comparisons to other work: Although I am not an expert in this particular area of model compression, the intuitive comparisons and mentions of prior work felt satisfying and informative. This problem is definitely timely, and the paper showed results of improved inference speed and memory footprint on both a smaller 10-way classification task as well as a larger 1000-way one.

My current decision is a weak accept, for a well-written paper and an interesting problem, but in the presence of some concerns and suggestions as listed below.

(1) My main concern is that without weight regularization, this optimization problem seems ill-posed. Looking at algorithm 1 specifically, consider the case where j=10 and the learned alphas are indeed sparse, like (0.1, 0, 0,…, 0, 0.9). The cost of J=0 here might be very high (i.e., large, slow, expensive model) and cost of J=9 might be very low (i.e., small fully-connected network that basically maps to an identity function). Here, the cost during optimization could be very low, and the results could be very accurate, but the weights of J=0 may be very high such that the entire work of getting the right answer is from this model. More generally, a low alpha can be attained for any model by simply increasing the network weights themselves. This doesn’t seem to be taken care of anywhere in this paper; if not addressed in any way, it would be a large reason for me to suggest rejecting this work.

(2) For the experiments, both of the chosen tasks were classification. Although one was smaller/easier and one was larger/harder, it would serve to be much more convincing if a task requiring regression and/or higher-dimensional output were tested. For these classification tasks, I can imagine that for certain model changes, the decision boundaries may remain the same even if the network itself is changing in detrimental ways. Thus, a task such as an image-to-image depth perception task (or other tasks in this category) would solidify the results and make it more convincing that DARC does indeed help.

Minor:

(a) The motivation of going from L0 to L1 for the cost constraint seems relevant also to the choice of h being the convex combination of different models. Can the motivation and problem setup address these motivations together, instead of separately?

(b) Although this paper defined R(h) and it also defined h=(\sum (alpha_j * h_j), it would still be helpful for the reader if the paper could explicitly define R(\sum (alpha_j * h_j)) somewhere. Although I know what was intended was to compose these h_j functions as sequential stages to pass data through, the sum notation seems a bit misleading like you may be averaging the final predictions/outputs from different models and computing the loss on these averaged results.


**Experience Assessment:**

I have read many papers in this area.

**Review Assessment: Checking Correctness Of Derivations And Theory:**

I assessed the sensibility of the derivations and theory.

**Review Assessment: Checking Correctness Of Experiments:**

I assessed the sensibility of the experiments.

**Review Assessment: Thoroughness In Paper Reading:**

I read the paper at least twice and used my best judgement in assessing the paper.

---

> ### Author Response · Authors · 2019-11-15
> **Author Response to Official Blind Review #3**
>
> Main Comments:
>
> Thank you for your comments and careful review of the paper. We respond to each of your comments below:
>
> (1) "[W]ithout weight regularization, this optimization problem seems ill-posed... More generally, a low alpha can be attained for any model by simply increasing the network weights themselves."
>
> Response: We thank the reviewer for this insightful question. We did in fact use weight regularization, and indeed the problem is ill-posed without this. The reason we did not mention this in the experiments is that, in MXNet, a weight decay parameter (of 10^{-4}) is built into stochastic gradient descent by default, and we did not modify this. We will add a note under "DARC Training Details" on page 5. In the theoretical formulation in Sections 2 and 3, the candidate classes \cal{H}_j can be assumed to have bounded L2 norm. As the reviewer noted, this plays an important role in ensuring that the L1 optimization problem is well-posed, so we will add a sentence to Section 2 clarifying this.
>
> (2) "For the experiments, both of the chosen tasks were classification… It would serve to be much more convincing if a task requiring regression and/or higher-dimensional output were tested."
>
> Response: The vast majority of prior work on neural network compression (including, e.g., MobileNets, MNasNet, FBNet, ProxylessNAS, etc.) has focused on image classification, and the existence of these baselines for comparison was a major reason for our focus on this task. However, we agree that applying DARC to neural networks for other tasks is a fruitful direction for future work.
>
> Minor Comments:
>
> (a) The motivation of going from L0 to L1 for the cost constraint seems relevant also to the choice of h being the convex combination of different models. Can the motivation and problem setup address these motivations together, instead of separately?
>
> Response: Thank you for this suggestion. Indeed, both the L1 cost and the convex combination are essentially artifacts of approximating a computationally challenging combinatorial optimization problem with a more tractable continuous relaxation. We alluded to this in the first paragraph of Section 2, but we can add more detail to make this more concrete.
>
> (b) Although this paper defined R(h) and it also defined h=(\sum (alpha_j * h_j), it would still be helpful for the reader if the paper could explicitly define R(\sum (alpha_j * h_j)) somewhere. Although I know what was intended was to compose these h_j functions as sequential stages to pass data through, the sum notation seems a bit misleading like you may be averaging the final predictions/outputs from different models and computing the loss on these averaged results.
>
> Response: The notation \sum (alpha_j * h_j) is actually meant to denote weighted averaging (convex combination) of the outputs of each candidate. To clarify, the candidates h_1,...,h_J discussed in Section 2 are for a single layer of the network. During training, DARC replaces each layer of the network with a convex combination of this form. This extension to multi-layer networks (compositions of layers) is discussed in Section 3 (specifically, Eq. (2)). We note that these intermediate layers of the network always output real-valued vectors (for classification, discretization is performed by a softmax layer at the output of the network, as usual in deep networks for classification), so this averaging is well-defined.
>
> The definition of R(h) should apply for the risk R(\sum (alpha_j * h_j)) of such a convex combination unchanged. We could, however, expand the formula for R(\sum (alpha_j * h_j)), if the reviewer feels this would help readers.

---

### Decision · Program_Chairs · 2019-12-19

**Decision:**

Reject

**Comment:**

The paper is on the borderline. A rejection is proposed due to the percentage limitation of ICLR.